# Erianin-Loaded Photo-Responsive Dendrimer Mesoporous Silica Nanoparticles: Exploration of a Psoriasis Treatment Method

**DOI:** 10.3390/molecules27196328

**Published:** 2022-09-26

**Authors:** Huanan Yu, Yuanqi Liu, Fang Zheng, Wenyu Chen, Kun Wei

**Affiliations:** School of Bioscience and Bioengineering, South China University of Technology, Guangzhou 510006, China

**Keywords:** erianin, mesoporous silica nanoparticles, photo-responsive, psoriasis, drug delivery

## Abstract

Psoriasis is a chronic inflammatory skin disorder accompanied by excessive keratinocyte proliferation. Erianin (Eri) is an ideal drug candidate for inhibiting proliferation and inducing apoptosis in the treatment of psoriasis. However, Eri’s poor water solubility and low penetration activity across the skin hinder its application in local medicine. In this study, we developed a novel photo-responsive dendritic mesoporous silica nanoparticle-based carrier to deliver erianin, improved its bioavailability, and achieved sustained-release effects. Spiropyran (SP), 3-aminopropyltriethoxysilane (APTES), and perfluorodecyltriethoxysilane (PFDTES) were conjugated to the outer surface, which allowed Eri to be released in response to UV radiation. The physicochemical properties of photo-responsive dendritic mesoporous silica nanoparticles (Eri-DMSN@FSP) were characterized via multiple techniques, such as using a Fourier-transform infrared spectrometer, a high-resolution transmission electron microscope, and nuclear magnetic resonance (NMR) spectroscopy. The anti-proliferative properties and light-triggered release of erianin-loaded photo-responsive dendritic mesoporous silica nanoparticles were assessed via the MTT assay and a drug release study in vitro. Erianin-loaded photo-responsive dendritic mesoporous silica nanoparticles (UV) exhibit a significantly enhanced HaCat cell-inhibiting efficacy compared to other formulations, as demonstrated by their extremely low cell viability of 10.0% (concentration: 500 mg/mL), indicating their capability to release a drug that responds to UV radiation. The cellular uptake of photo-responsive dendritic mesoporous silica nanoparticles (DMSN@FSP) was observed via confocal laser scanning microscopy (CLSM). These experimental results show that Eri-DMSN@FSP could be effectively endocytosed into cells and respond to ultraviolet light to release Eri, achieving a more effective psoriasis treatment. Therefore, this drug delivery system may be a promising strategy for addressing the question of Eri’s delivery and psoriasis therapy.

## 1. Introduction

Psoriasis is a common chronic inflammatory skin disorder that affects people of all ages and poses a tremendous burden for individuals and society [1,2]. Typical lesions are monomorphic, sharply demarcated erythematous plaques covered by silvery lamellar scales caused by excess keratinocyte proliferation and immune cell infiltration [3,4]. The skin redness is because of increased numbers of tortuous capillaries that reach the skin surface through a markedly thinned epithelium. The inflammatory infiltrate consists primarily of dendritic cells, macrophages, T cells, neutrophils in the dermis, and some T cells in the epidermis [5]. In addition, the tumor necrosis factor (TNF)-α and interferon (IFN)-γ in keratinocytes activate the inflammatory response via stimulating the secretion and synthesis of various inflammatory mediators [6]. Psoriasis can be induced via various factors in genetically susceptible individuals, including trauma, infection, and medications (such as β-blockers, IFNα, and lithium) [7]. 

In current psoriasis treatments, chemotherapy is still a typically utilized option [8]. However, traditional chemotherapy agents (such as methotrexate, corticosteroids, and calcineurin inhibitors) cause significant side effects on the body. Moreover, the emergence of drug resistance has attenuated the effectiveness of chemotherapy [9]. Although biologics are effective, their high cost limits their use [10]. Therefore, a more effective, less toxic, and low-cost therapeutic strategy is urgently needed for psoriasis treatment. 

Currently, natural products have served as inspiration for scientists, both for their complex three-dimensional structure and exquisite biological activity [11,12]. Erianin (Eri) is a low-molecular-weight natural product named 2-methoxy-5-[2-(3,4,5-trimethoxy-phenyl)- ethyl]-phenol, extracted from Dendrobium chrysotoxum Lindl [13]. Recent studies have shown that Eri is also capable of inhibiting proliferation and inducing apoptosis in HaCat, a spontaneously immortalized human keratinocyte previously used as a psoriasis model [14]. These results imply that Eri may be an effective clinical treatment strategy for psoriasis. However, its poor water solubility and low penetration activity across the skin (stratum corneum) have limited its topical application [15]. Therefore, the development of a drug delivery system that can be applied topically and deliver Eri precisely to its target is highly desirable.

Recent advances in biomedical nanotechnology have led to the development of multifarious nanomaterial-based drug delivery systems (DDSs) [16]. For example, various nanoparticle-based drug delivery systems have been developed and used in cancer therapy in recent years [17,18]. Mesoporous silica nanoparticles, among many different nanoparticles, have attracted considerable attention for drug delivery because of their exceptional morphological characteristics, including their large surface area-to-volume ratio and pore volume, the ability to tailor shape, diameter and porosity, and the ease of abundant surface chemistry, along with their extraordinary physicochemical and biological properties [19,20]. In light of these explicit characteristics, MSNs have been proven to be promising nanocarriers that have revolutionized different approaches to drug delivery [21], such as controlled [22], targeted [23], sustained [24], and responsive systems [25,26]. In addition, they exhibit excellent biocompatibility, although they degrade slowly [27]. These paramount features make MSNs excellent nanoplatforms to load erianin for psoriasis treatment [28,29,30]. Furthermore, recent studies have shown that Eri can be delivered by dendritic mesoporous silica nanoparticles (DMSNs), which can improve its bioavailability [14]. However, Eri-loaded DMSNs have some disadvantages, such as a lack of controlled-release and sustained-release capabilities, which makes blood concentration fluctuate greatly and increases the side effects of the drug. In order to reduce the side effects of the drug, the device should be modified to slow down or control the release of the drug.

In this study, we developed Eri-DMSN@FSP (FSP represents spiropyran (SP) and perfluorodecyltriethoxysilane (PFDTES), two chemical reagents), a UV-responsive erianin-loaded dendritic mesoporous silica nanoparticle. Furthermore, we demonstrated that Eri-DMSN@FSP was able to deliver Eri into HaCat cells in a controlled and sustained manner. Our study also showed that Eri-DMSN@FSP significantly inhibited proliferation and promoted apoptosis for psoriasis cellular models.

## 2. Results and Discussion

### 2.1. Characterization

The synthesis process of DMSN@FSP is illustrated in Figure 1a. The main drug-release mechanism of photo-responsive DMSNs is drug diffusion through water-filled pores. Spiropyran (SP) is a photosensitive molecule with switchable properties that have been extensively studied [31]. When irradiated with ultraviolet light with a wavelength of 365 nm, the spiropyran molecule changes from a hydrophobic state into a hydrophilic state, which causes the wetting of the nanoparticles. Then, the drug is released from the pores [32]. The chemical structures of spiropyran and erianin are shown in Figure 1b,c. The transmission electron microscopy (TEM) images show that DMSN, DMSN- FSP, and Eri-DMSN@FSP were highly dispersed, with a uniform particle size of 98–130 nm (Figure 2). Compared with the DMSN image, the deep color of the DMSN@FSP image indicates that the hydrophobic layer not only covered the DMSN outer surface, but also adhered to the inner surfaces of the channels (Figure 2a,b). The TEM image of Eri-DMSN@FSP shows no obvious morphology and dispersibility change after loading erianin (Figure 2c). The nitrogen adsorption–desorption results of DMSN@FSP are shown in Figure 3. They have a high Brunauer–Emmett–Teller (BET) surface area of 862 m^2^g^−1^ and a large pore volume of 3.19 cm^3^g^−1^. The well-connected mesopores are favorable for efficient drug loading, and the large pore volume can guarantee a large drug-loading capacity. The BJH pore size distribution determined from the adsorption branch shows a peak centered at 3.8 nm. The loading efficiency of Eri-DMSN@FSP is 71.57 ± 0.33%, determined by HPLC analysis. 

The hydrodynamic diameters and zeta potentials of DMSN and DMSN@FSP were measured by dynamic light scattering (DLS) and the results are shown in Figure 4. The average hydrodynamic diameter of the synthesized DMSNs was 101.4 nm, with a PDI of 0.017 and zeta potential of −27.6 mV, whilst that of the DMSN@FSPs was 104.2 nm, with a PDI of 0.024 and zeta potential of +12.1 mV. The diameter of the DMSN@FSPs was slightly larger than that of the DMSNs, indicating a successful surface modification. The negatively charged surface before functionalization was attributed to the partially hydrolyzed silanol groups. The change in zeta potential from −27.6 mV to +12.1 mV was due to the positive charge of the SP [33]. The structures of SP-COOH and DMSN@FSP were characterized by ^1^H-NMR and solid-state ^13^C-NMR, respectively. The active hydrogen of the carboxyl group was rapidly exchanged with the active hydrogen in the solution, so it was not visible (Figure 5a). The a and b peaks of ^13^C-NMR spectroscopy were assigned to the ester carbonyl carbon, indicating that DMSN@FSP was successfully synthesized [32] (Figure 5b). The Fourier-transform infrared (FTIR) technique was used to investigate the molecular bond signatures of DMSN and DMSN@FSP (Figure 6a). In the FTIR spectra, the black arrows indicate the characteristic FTIR absorbance peaks of the nanoparticles. In the spectrum of DMSN, 960 cm^−1^ and 802 cm^−1^ are attributed to the bending vibration of the silanol group and the asymmetric stretching vibration of the silicon-oxygen bond, respectively [34]. In the FTIR spectrum of DMSN@FSP, the absorption bands at 1635 and 1728 cm^−1^ were assigned to the vibration of the amide and ester groups, respectively, indicating the successful attachment of spiropyran to DMSN@FSP. Moreover, the absorption peaks at 2850 and 2917 cm^−1^ are the stretching vibration peaks of methylene. The successful DMSN@FSP modification was also verified by the X-ray photoelectron spectroscopy (XPS) studies. As shown in Figure 6b, the full XPS survey spectrum showed the existence of F.

Thermogravimetric analysis (TGA) was used to determine the drug-loading capacity and thermal stabilities of the nanoparticles (Figure 7a). When the temperature reached 200 °C, the mass loss caused by water evaporation in the pores of DMSN@FSP was about 3.62%, and Eri-DMSN@FSP had almost no mass loss at that time. When the temperature reached 400 °C, the mass loss of Eri was close to 100%, which indicated that Eri would completely vaporize at 400 °C. When the temperature reached 425 °C, the mass loss of Eri-DMSN@FSP was 44.82%. The results indicated that the protective encapsulation by DMSN@FSP improved the thermal stability of erianin and effectively prevented its rapid decomposition at high temperatures. Figure 7b shows a differential scanning calorimetry (DSC) curve of Eri, DMSN, DMSN@FSP, and Eri-DMSN@FSP. Eri shows a sharp endothermic peak at about 100 °C, implying its crystalline structure [35]. A weak peak was observed in Eri-DMSN@FSP, indicating that Eri existed in the channels in a non-crystalline state, thus confirming the successful Eri loading into the nanopores of DMSN@FSP [36].

### 2.2. In Vitro Drug Release Study

The release study of Eri, Eri-DMSN, Eri-DMSN@FSP, and Eri-DMSN@FSP (UV) was conducted using a PBS solution as the medium. As shown in Figure 8, the release profile of Eri experienced an initial burst release followed by a sustained release. More than 38% of pure erianin was released within 5 h, and another approximately 62% was released in the next 43 h. In comparison, the release rate of Eri was slightly slower in DMSN, and almost no Eri was released in DMSN@FSP. The Eri release rate was markedly slowed down after Eri was loaded into DMSN@FSP, suggesting that the mesopores and channels were covered and blocked by the immobilized decoration molecules. When the UV light (365 nm) was turned on, the release rate of Eri became faster from Eri-DMSN@FSP, which demonstrated that the release responds to ultraviolet light. 

### 2.3. In Vitro Cytotoxicity Assays 

Taking the blank nanocarrier DMSN@FSP as a blank control, its biocompatibility was explored with the MTT experiment (Figure 9a). The results showed that the blank DMSN@FSP had almost no toxicity after co-culture with HaCat cells for 24 h, proving the good biocompatibility of DMSN@FSP. Then, the in vitro treatment effect of Eri-DMSN@FSP was investigated using an MTT assay. As shown in Figure 9b, with an increased concentration of Eri, Eri-DMSN, Eri-DMSN@FSP and Eri-DMSN@FSP (UV), the cell viability started to gradually drop, which suggests they have a significant effect on the viability of HaCat cells. The cytotoxicity of free Eri to HaCat cells was decreased (cell viability increased) significantly by loading into DMSN and DMSN@FSP carriers. The controlled-release capacity of the carriers, resulting in decreased Eri concentrations, could be responsible for their decreased cytotoxicities. Eri-DMSN@FSP (UV) exhibited significantly enhanced HaCat-cell-inhibiting efficacy compared to other formulations, as demonstrated by its extremely low cell viability of 10.0% (concentration: 500 μg/mL), proving it possesses the capacity to release a drug that responds to UV radiation.

### 2.4. Cellular Uptake and Intracellular Distribution of Eri-DMSN@FSPs

To assess DMSN@FSP as a drug delivery vehicle, intracellular uptake experiments were carried out using HaCat cells (Figure 10). FITC was loaded into DMSN@FSP as a fluorescent probe. In the FITC group, the intensity of the green fluorescence signal gradually increased in a time-dependent manner, indicating that DMSN@FSP was gradually taken up by the cells. After 24 h, most of the nuclei of the DAPI group were stained blue, and the blue nuclei of the merged group were encapsulated by green nanoparticles, indicating that DMSN@FSP was able to be readily internalized into HaCat cells [37]. 

## 3. Materials and Methods

### 3.1. Materials 

Tetraethyl orthosilicate (TEOS), 1-octadecene (ODE), cetyltrimethylammonium chloride (CTAC) solution (25 wt % in H_2_O), fluorescein isothiocyanate (FITC), and triethanolamine (TEA) were purchased from Aladdin Reagent (Shanghai, China). Succinic anhydride, 1-(2-Hydroxyethyl)-3,3-dimethylindolino6’-nitrobenzopyrylospiran (SP), dichloromethane, triethylamine, and 4-dimethylaminopyridine were purchased from Macklin Biochemical (Shanghai, China). Anhydrous toluene, 3-aminopropyltriethoxysilane (APTES), and 1H,1H,2H,2HPerfluorodecyltriethoxysilane were purchased from Sigma-Aldrich (PFDTES). N-(3-Dimethylaminopropyl)-N′- ethylcarbodiimide hydrochloride (EDC), N-Hydroxysuccinimide(NHS), and anhydrous ethanol were obtained from Rhawn Reagent. Erianin (≥98% purity) was obtained from Chengdu Herbpurify (Chengdu, China). HaCaT cells CAS were obtained from Kunming Cell Bank (Yunan, China). Dulbecco’s Modified Eagle’s Medium (DMEM) and fetal bovine serum were obtained from Wisent Corporation (Nanjing, China). Penicillin streptomycin was obtained from Thermo Fisher Scientific (Waltham, MA, USA). Phosphate-buffered saline (PBS) was obtained from Procell (Wuhan, China). MTT reagents were purchased from Biofroxx (Einhausen, Germany). Dimethyl sulfoxide (DMSO) was obtained from MP Biomedicals (Irvine, CA, USA). A dialysis bag was provided from Shanghai Yuan Ju Biological Technology. Deionized water was used in all experiments.

### 3.2. Preparation of DMSN@FSPs

#### 3.2.1. Synthesis of DMSNs

The uniform DMSN was synthesized via the surfactant-directed self-assembly method [38]. First, 48 mL of cetyltrimethylammonium chloride solution, 72 mL of deionized water, and 0.36 g of triethanolamine were added into a 250 mL round-bottom flask in a 60 °C water bath with gentle magnetic stirring for 1 h. Second, 20 mL of the 1- octadecene solution, including 20 *v*/*v*% tetraethyl orthosilicate, was slowly added to the reactant and kept for another 17 h. The sample was collected by centrifuging the aqueous solution (8000 rpm) and was washed three times with anhydrous ethanol. Subsequently, the product was vacuum-dried at 50 °C for 12 h. Finally, the obtained DMSN was treated by high-temperature calcination at 550 °C for 5 h to remove the template.

#### 3.2.2. Synthesis of DMSN-FNH_2_

The DMSN was alternately modified with APTES and PFDTES as the sources of amino groups and hydrophobic dopants, respectively, to form DMSN-FNH_2_. Amounts of 0.4 mL of APTES and 0.8 mL of PFDTES were slowly added to 25 mL of dry toluene. An amount of 3 g of as-prepared DMSN was dispersed in the solution by ultrasound. Afterwards, the mixture solution was heated to 110 °C and stirred at this temperature for 24 h. After the reaction, the resultant solid was separated by centrifugation, followed by washing with ethanol and drying overnight. The next day, the DMSN-FNH2 white powder was obtained.

#### 3.2.3. Synthesis of SP-COOH

SP-COOH was constructed according to the methods documented in the existing literature [39,40]. First of all, 1.6 g of SP was added into 120 mL of a dichloromethane solution containing 570 mg of succinic anhydride, 200 mg of 4-dimethylamino pyridine, and 3 mL of triethylamine. The mixture was stirred for 24 h at room temperature, followed by the solvent being removed under vacuum. The collected product was purified by gel chromatography eluted with a 50% acetic ester to petroleum ether eluent to produce a pink powder of SP-COOH.

#### 3.2.4. Preparation of DMSN@FSP

DMSN@FSP was achieved through DMSN-FNH_2_ conjugation with SP-COOH, and the synthesis of the carboxyl group was introduced into SP. The synthesized DMSN-FNH_2_ was dispersed by sonication into an ethanol solution containing 200 mg of SP-COOH, 580 mg of EDC, and 400 mg of NHS. The mixture was stirred at room temperature for 20 h, and the solution was then centrifuged and washed with deionized water several times. The samples were dried at 50 °C under vacuum for 24 h. Finally, DMSN-FSP was successfully prepared.

### 3.3. Drug Loading Procedure

Eri was loaded into DMSN@FSP via the rotary evaporation technique, with slight modifications to the method in the literature [41]. An amount of 150 mg of Eri was dissolved in 5 mL of dichloromethane and ultrasonicated for 20 min to promote dissolution, and a 30 mg/mL erianin solution was obtained. An amount of 250 mg DMSN@FSP was dispersed in the solution by sonication. Afterwards, the solution was transferred to a rotary evaporation flask. The solvent was slowly evaporated using a rotary evaporator at 35 °C until all the solvent was removed and a dry powder was observed in the flask. Finally, the sample was collected and further dried in a vacuum oven. Eri-DMSN@FSP was successfully prepared. All reactions were protected from light.

### 3.4. Characterization

The size distribution and zeta potential of nanoparticles were measured by dynamic light scattering (DLS, Nano-Zetasizer, Malvern, UK). Each measurement was performed at 20 °C and repeated at least three times. The morphological examination was carried out with transmission electron microscopy (TEM) (H-7650, HITACHI Ltd., Tokyo, Japan). The pore size, pore volume, and surface area were analyzed using N 2 adsorption/desorption isotherms via an Autosorb-iQ automated gas sorption analyzer (Quantachrome, Boynton Beach, FL, USA). Thermogravimetric analysis (TGA) and differential scanning calorimetry (DSC) analysis were performed using a simultaneous thermal analyzer (STA 449C, Netzsch Ltd., Weimar, Germany). Four analysis samples were placed in aluminum pans, and the TGA experiment was carried out from 25 to 800 °C at a heating rate of 10 °C/min. The Fourier-transform infrared (FTIR) spectra were obtained using an iS10 FT-IR spectrometer (Thermo Nicolet Corporation, Waltham, MA, USA). The 1 H NMR, 13 C NMR, and solid-state 13 C NMR spectra were obtained using a high-resolution NMR spectrometer at 600 MHz, (Bruker AVANCE III 600 M, Ettlingen, Germany). The X-ray photoelectron spectrum (XPS) was performed on an Axis Ultra DLD (KratOs, Manchester, UK).

### 3.5. Analysis of Loading Efficiency

Freshly prepared 10 mg of Eri-DMSN@FSP was dispersed in 1 mL of methanol by sonicating for 20 min. Subsequently, the mixture was centrifuged and filtered with a 0.22 μm pore size membrane filter for the HPLC analysis. To perform the chromatographic separation, we used an Agilent C18 column (4.6 mm × 250 mm, 5 μm) at 35 °C. The mobile phase was acetonitrile and water (volume ratio is 30 to 70), and was eluted in a gradient mode at a flow rate of 1.0 mL/min. The detection wavelength was set at 232 nm [14]. The DMSN@FSP loading capacity was calculated using the following equations: loading efficiency (%) = (amount of encapsulated Eri/Total amount of the complex) × 100%

### 3.6. In Vitro Drug Release Study

The in vitro release study of Eri from each of the drug-loaded carriers was performed in phosphate-buffered saline (PBS, pH 7.4) at 37 °C to mimic physiological and skin temperature [42,43]. Each of the aqueous dispersions (5 mL, 1 mg/mL) of Eri, Eri-DMSN, Eri-DMSN@FSP, and EriDMSN@FSP (UV) were sealed in four dialysis bags (8000–14,000 molecular weight cut-off) and immersed in different centrifuge tubes containing 45 mL of phosphate-buffered saline (PBS) at pH 7.4, respectively. The centrifuge tubes were transferred to an orbital shaker set to 80 rpm. A 3 mL aliquot of the dialysate was retrieved for analysis and replenished with 3 mL of its fresh dialysate at predetermined intervals (1, 2, 3, 4, 5, 6, 7, 8, 11, 14, 24, 36, 48 h). The Eri concentration of each collected sample was determined using HPLC. The cumulative release rate was calculated according to our previous report.

### 3.7. Cell Culture

Immortalized human epidermal keratinocytes (HaCat) cells were cultured in DMEM, supplemented with 10% FBS, 1% penicillin-streptomycin (100 U/mL penicillin and 100 mg/mL streptomycin) in a humidified incubator at 37 °C in a 5% CO_2_ humidified atmosphere.

### 3.8. In Vitro Cytotoxicity Assay

The in vitro cytotoxicities of the blank DMSN, blank DMSN@FSP, Eri, and Eri-loaded DMSN@FSP were evaluated by the MTT assay, according to a previous publication [44,45]. HaCat cells, spontaneously immortalized in human keratinocyte line, are often used for the study of psoriasis [46]. Therefore, the cell experiments in this study were carried out on HaCat cells. Briefly, the HaCat cells were plated at a density of 5 × 105 cells per well in 96-well plates and incubated at 37 °C with 5% CO_2_. After starvation in a serum-free medium for 24 h, the medium was removed and replaced with 100 μL of DMEM containing different concentrations of Eri, blank DMSN, blank DMSN@FSP, and Eri-DMSN@FSP. The blank control group was cultured in the same conditions without nanoparticles and drug treatments. After 24 h, cells were then treated with the MTT reagent (20 μL/well) for 4 h at 37 °C. Afterwards, 200 μL of DMSO was added to each well. The optical density (OD) was measured at 570 nm using a microplate reader and the percentage of cell viability was determined.

### 3.9. Cellular Uptake and Intracellular Distribution of Eri-DMSN@FSPs

The cellular uptake and intracellular distribution of Eri-DMSN@FSP were investigated according to a reported method, with a slight modification [47]. Briefly, HaCat cells were seeded on the coverslip in 24-well plates at 1 × 10^5^ cells per well and grown overnight. Then, the medium was removed and replaced with fresh DMEM containing FITC-labeled DMSN@FSP (200 mg/mL), and subsequently cultured for 2 h, 4 h, 8 h, and 24 h. The cells were washed twice with PBS to remove the remaining nanoparticles. Slides were fixed with 4% paraformaldehyde (PFA) and treated with 0.3% Triton-X 100. The cell nuclei were distinguished by staining with blue fluorescent DAPI (4, 6-diamidino-2-phenylindole) at room temperature for 10 min, after which they were washed and imaged.

### 3.10. Statistical Analysis

All experiments were performed in at least three independent experiments. Origin2019b software was used for the statistical analysis. The results are presented as mean ± SD. * *p* < 0.05, ** *p* < 0.01, *** *p* < 0.001 was considered statistically significant.

## 4. Conclusions

In this study, we developed a novel photo-responsive dendritic mesoporous silica nanoparticle-based carrier to deliver erianin, improved its bioavailability, and achieved sustained-release effects. TEM showed that blank DMSN@FSP nanoparticles are spherical nanoparticles composed of dendritic channels with a uniform particle size of 98–130 nm. DLS demonstrated the good stability of DMSN@FSP in the PBS. The results of the BET analysis showed that DMSN@FSP has a large drug-carrying capacity. The in vitro release and cell experiments demonstrated that Eri-DMSN@FSP displays the capacity to modulate drug release using UV light as the external stimuli. Eri-DMSN@FSP significantly optimizes erianin’s release rate, and thus will be capable of reducing fluctuations in drug concentration in blood and side effects of the drug. In conclusion, the light-responsive dendritic mesoporous silica nanoparticles loaded with erianin constructed in this study may offer a new strategy for psoriasis treatment. However, this study focused solely on the cellular effects of Eri-DMSN@FSP on psoriasis, and the follow-up study will investigate its efficacy in animals. 

## Figures and Tables

**Figure 1 molecules-27-06328-f001:**
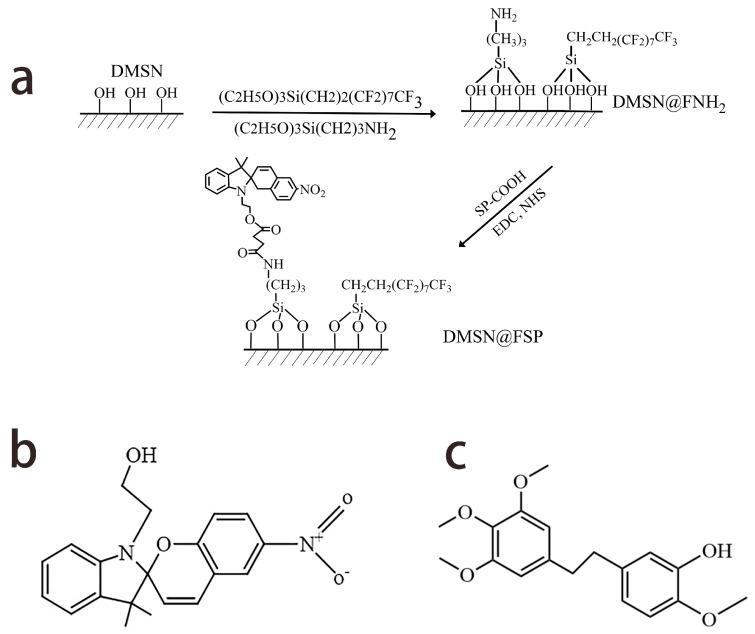
(**a**) The synthesis process of DMSN@FSP. (**b**) The chemical structures of spiropyran. (**c**) The chemical structures of erianin.

**Figure 2 molecules-27-06328-f002:**
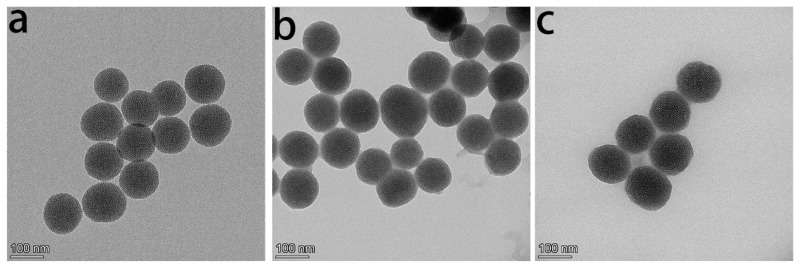
TEM images of DMSN (**a**), DMSN@FSP, (**b**) and Eri-DMSN@FSP (**c**).

**Figure 3 molecules-27-06328-f003:**
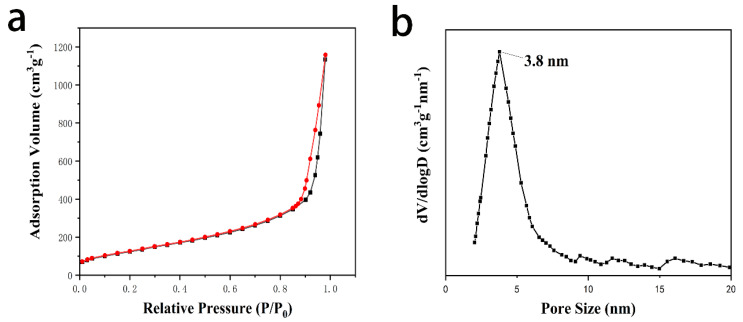
(**a**) Nitrogen sorption isotherms; (**b**) BJH pore size distribution curve derived from the adsorption branches of DMSN.

**Figure 4 molecules-27-06328-f004:**
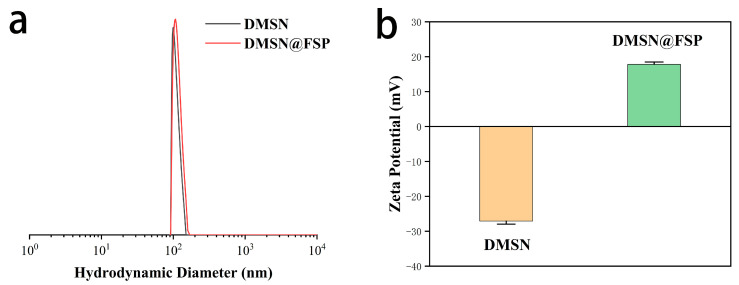
DLS data. (**a**) Particle size distribution of DMSN and DMSN@FSP; (**b**) Zeta potential of DMSN and DMSN@FSP.

**Figure 5 molecules-27-06328-f005:**
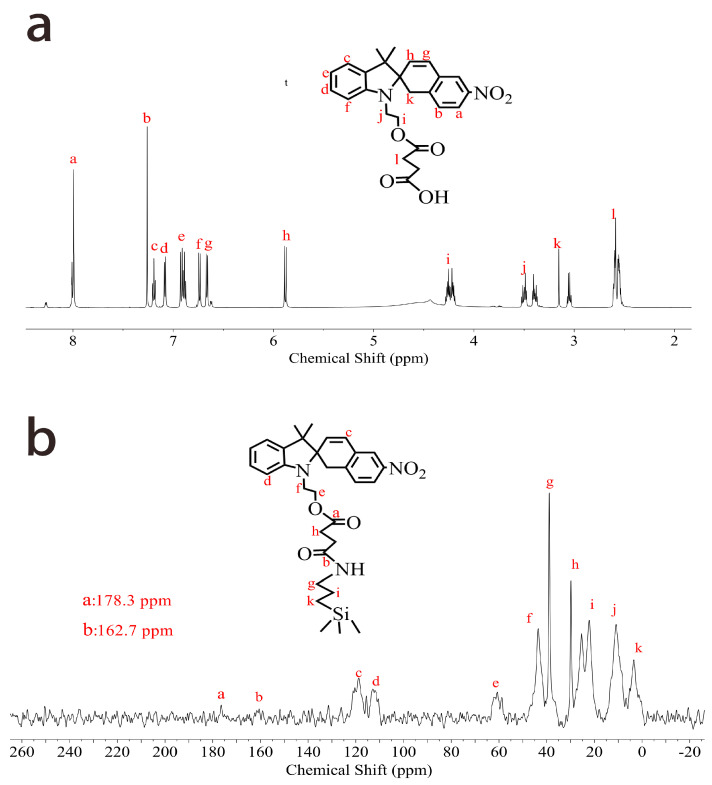
(**a**) ^1^H NMR spectroscopy of SP−COOH; (**b**) solid-state ^13^C NMR spectroscopy of DMSN@FSP.

**Figure 6 molecules-27-06328-f006:**
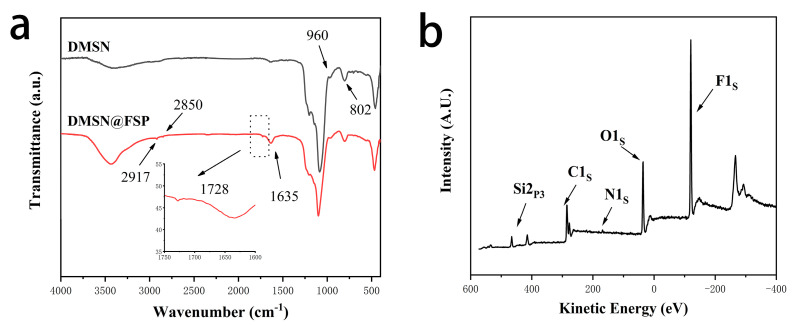
(**a**) Fourier−transform infrared spectroscopy of DMSN and DMSN@FSP. (**b**) Results of the XPS analysis of DMSN@FSP.

**Figure 7 molecules-27-06328-f007:**
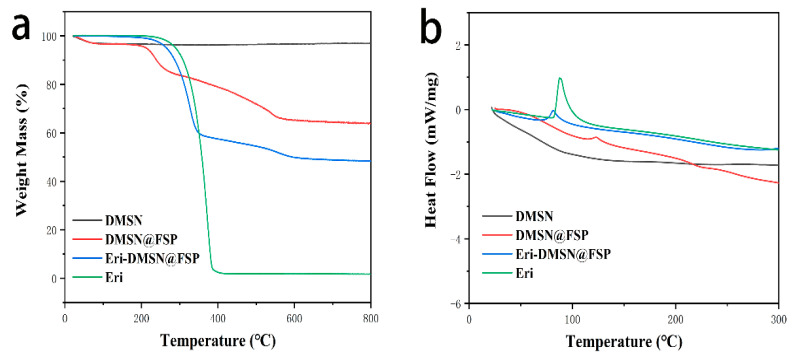
(**a**) Thermogravimetric analysis (TGA) curves of DMSN, DMSN@FSP, Eri−DMSN@FSP, and Eri. (**b**) Differential scanning colorimetry (DSC) profile of DMSN, DMSN@FSP, Eri−DMSN@FSP, and Eri.

**Figure 8 molecules-27-06328-f008:**
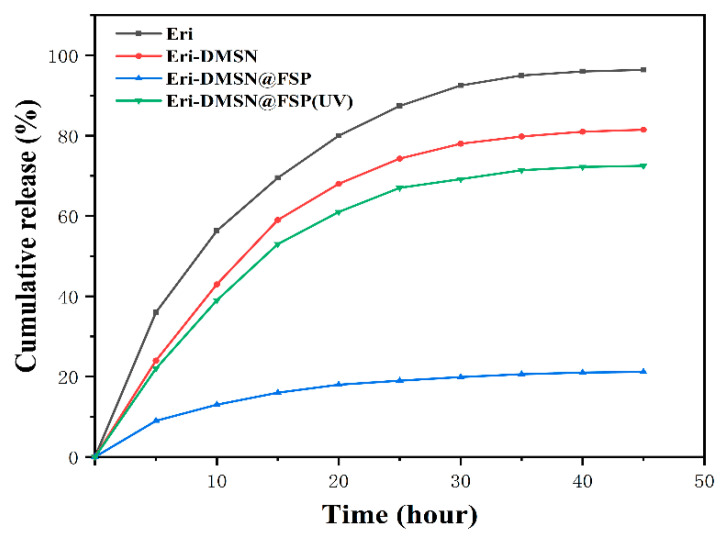
The in vitro release curves of Eri, Eri-DMSN, Eri-DMSN@FSP, and Eri-DMSN@FSP(UV) in PBS (pH 7.4).

**Figure 9 molecules-27-06328-f009:**
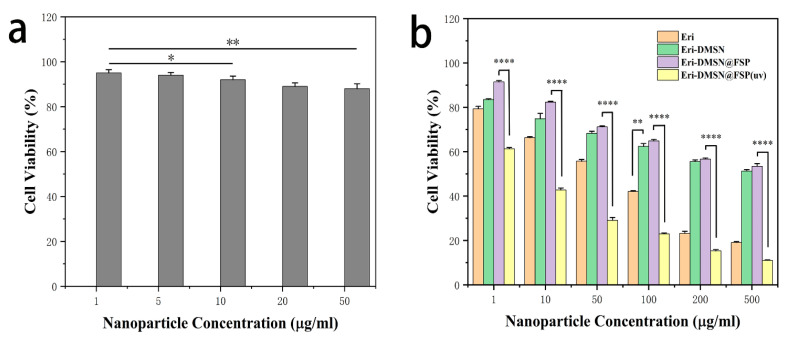
(**a**) In vitro cytotoxicity study of blank DMSN@FSPs after 24 h treatment (*n* = 3, mean ± SD). (**b**) Cell viability of HaCat cells subject to treatment of Eri, Eri-DMSN, Eri-DMSN@FSP, and Eri-DMSN@FSP(UV) for 48 h (*n* = 3, mean ± SD), from three independent experiments. * *p* < 0.05, ** *p* < 0.01, **** *p* < 0.0001.

**Figure 10 molecules-27-06328-f010:**
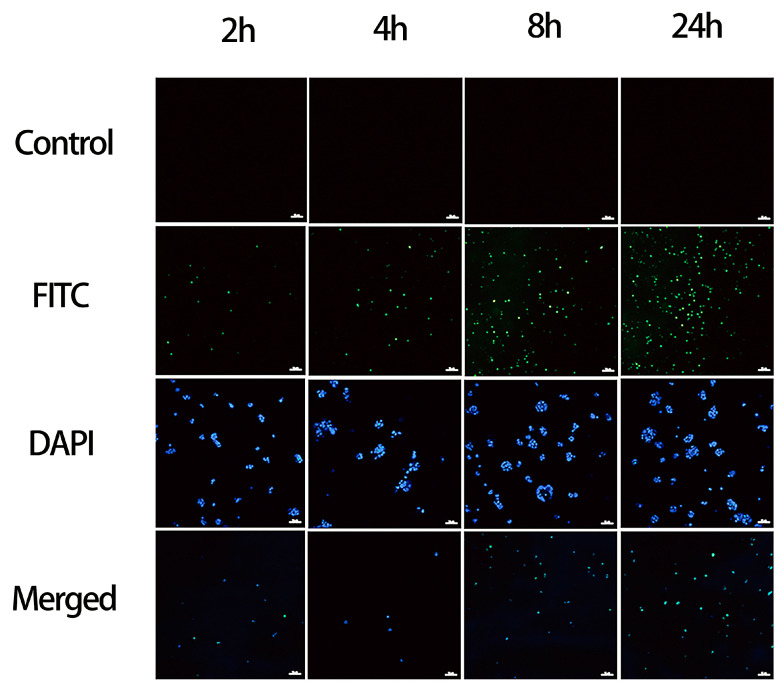
CLSM images of HaCat cells cultured in fresh DMEM, containing FITC-labeled DMSN@FSP for 2 h, 4 h, 8 h, and 24 h. The scale bars are 20 μm.

## Data Availability

Not applicable.

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
