# Peer review of "Erianin-Loaded Photo-Responsive Dendrimer Mesoporous Silica Nanoparticles: Exploration of a Psoriasis Treatment Method"

_molecules, 2022, doi:10.3390/molecules27196328_

Round 1
Reviewer 1 Report
The study is good its a nice initiative, because still psoriasis has to addressed. The authors have to improve the terms and language. hours is mentioned as "hours" and "h" in many places of the manuscript.
The references in introduction has to be increased that explain the drug loading and importance of nanoparticles. add these references
a) Quercetin loaded PLGA microspheres induce apoptosis in breast cancer cells, 2019, Applied Surface Science 487, 211-217
b) Role of mesoporous silica nanoparticles for the drug delivery applications , Materials Research Express Open Access Volume 7, Issue 10October 2020
c) Elicitor-Induced Metabolomics Analysis of Halodule pinifolia Suspension Culture for an Alternative Antifungal Screening Approach against Candida albicans , Journal of FungiOpen AccessVolume 8, Issue 6June 2022 Article number 609.
d)Skin Cancer Identification, Lecture Notes in Electrical Engineering Volume 690, Pages 501 - 5072021
2) Discussion part must have information on ,mechanism.
3) Check the english also
Reviewer 2 Report
The authors presented the paper "Erianin-loaded photoresponsive dendrimer mesoporous silica nanoparticles: Exploration of a psoriasis treatment method"
1) A bit more 2-3 years references recommended to use in the Introduction section to show the progress in the area. Moreover, I recommend to present some more information about silica nanoparticles advantages in general with 2-3 years review references. Abbreviation FSP is decrypted in the Introduction. It will be very good presenting the meaning.
2) The Abstract and Conclusions sections may be improved in the presenting of the paper novelty and quantitative results about the disease treatment. Moreover, for the Conclusion section it will good to write the limitations and further possible work development.
3) Figure 5A. I think there is a mistake in the spectrum interpretation, see the attachment (simulation in ChemDraw). Signal e can't at about 6 ppm and have d-peak 6-7 it is an aromatic place on the spectrum.
Figure 5B Other signals in the spectrum? Signals a and b is very low to decide about that it is COOH groups. What is the problem with baseline in 75-110 ppm. Smth. was deleted.
4) I see in the section 2.2 drug release study. But I haven't seen the drug-loading studies. What is your nanoparticles capacity of the drug?
5) Moreover, some comment should be mentioned why you have used pH 7.4. Where your drug will release in organism. It looks like in plasma. But have you studied the release using plasma or serum?
6) Why you have used HaCat cells? Is it good that you drug-loaded nanoparticles have 60% viability? Some discussion has to be added. After UV, it is ok.
Minor comments
Eri-DMSN@FSP abbreviation should be avoided in the abstract or explained.
Figure 1 mistake not -OH-Si, but -O-Si. The resulted organic molecule in the figure DMSN@FSP looks not very good. Moreover, there is smth. with angles and image resolution. The ChemDraw or ChemDraw online will be better.
Figure 4 resolution is not good, in the caption mention that a is DLS data
Figure 5 1H and 13C in superscript (the same in the text)
Figure 6 bad resolution of the insert picture
Figure 10 bad contrast

Round 2
Reviewer 2 Report
Thank you for the revised paper.